# Diverse Interactions of Sterols with Amyloid Precursor Protein Transmembrane Domain Can Shift Distribution Between Alternative Amyloid-β Production Cascades in Manner Dependent on Local Lipid Environment

**DOI:** 10.3390/ijms26020553

**Published:** 2025-01-10

**Authors:** Pavel E. Volynsky, Anatoly S. Urban, Konstantin V. Pavlov, Yaroslav V. Bershatsky, Olga V. Bocharova, Anastasia K. Kryuchkova, Veronika V. Zlobina, Alina A. Gavrilenkova, Sofya M. Dolotova, Anna V. Kamynina, Olga T. Zangieva, Amir Taldaev, Oleg V. Batishchev, Ivan S. Okhrimenko, Tatiana V. Rakitina, Roman G. Efremov, Eduard V. Bocharov

**Affiliations:** 1Shemyakin–Ovchinnikov Institute of Bioorganic Chemistry RAS, 117997 Moscow, Russia; volynski@yandex.ru (P.E.V.); anatoly.urban@gmail.com (A.S.U.); bershackyjaroslav@gmail.com (Y.V.B.); o.bocharova@gmail.com (O.V.B.); garaewa.nika@yandex.ru (V.V.Z.); alycat1008@gmail.com (A.A.G.); sm.dolotova@gmail.com (S.M.D.); aneskaminina@mail.ru (A.V.K.); r-efremov@yandex.ru (R.G.E.); 2Moscow Center of Advanced Studies, 123592 Moscow, Russia; qpavlov@mail.ru (K.V.P.); anastasiakruckova44@gmail.com (A.K.K.); t-amir@bk.ru (A.T.); ivan@nmr.ru (I.S.O.); 3Pirogov National Medical and Surgical Center, 105203 Moscow, Russia; zolgat@yandex.ru; 4Institute of Biomedical Chemistry, 119121 Moscow, Russia; 5Frumkin Institute of Physical Chemistry and Electrochemistry RAS, 119071 Moscow, Russia; olegbati@gmail.com; 6Department of Applied Mathematics, National Research University Higher School of Economics, 101000 Moscow, Russia

**Keywords:** Alzheimer’s disease, amyloid precursor protein, transmembrane domain, alternative cleavage cascades, lipid membrane state, cholesterol, NMR, molecular dynamics

## Abstract

Alzheimer’s disease (AD) pathogenesis is correlated with the membrane content of various lipid species, including cholesterol, whose interactions with amyloid precursor protein (APP) have been extensively explored. Amyloid-β peptides triggering AD are products of APP cleavage by secretases, which differ depending on the APP and secretase location relative to ordered or disordered membrane microdomains. We used high-resolution NMR to probe the interactions of the cholesterol analog with APP transmembrane domain in two membrane-mimicking systems resembling ordered or perturbed lipid environments (bicelles/micelles). In bicelles, spin-labeled sterol interacted with the peptide near the amphiphilic juxtamembrane region and N-terminal part of APP transmembrane helix, as described earlier for cholesterol. Upon transition into micellar environment, another interaction site appeared where sterol polar head was buried in the hydrophobic core near the hinge region. In MD simulations, sterol moved between three interaction sites, sliding along the polar groove formed by glycine residues composing the dimerization interfaces and flexible hinge of the APP transmembrane domain. Because the lipid environment modulates interactions, the role of lipids in the AD pathogenesis is defined by the state of the entire lipid subsystem rather than the effects of individual lipid species. Cholesterol can interplay with other lipids (polyunsaturated, gangliosides, etc.), determining the outcome of amyloid-β production cascades.

## 1. Introduction

The first indication of the close interrelation between Alzheimer’s disease (AD) and the imbalance of lipid homeostasis was reported by Alois Alzheimer himself, who included an increased rate of occurrence of adipose inclusions and lipoid granules into the list of the primary pathophysiological traits of the condition. The link between the pathogenesis of AD and the membrane lipids had faded into oblivion for a considerable time, overshadowed by the attention to the Aβ family peptides and other products of amyloid precursor protein (APP) cleavage by secretases, but the recognition of the strong and multifaceted interrelations between the composition of the lipid membranes and the disease occurrence and progression has been growing stronger lately. For example, one of the early APP cleavage products, the APP intracellular domain (AICD), has been reported to regulate, upon translocation into the nucleus, the transcription of genes encoding several enzymes involved in lipid metabolism (among many others) [1]. The referenced review by Grimm et al. provides a systematic and convincing evidence of interactions of various players in the AD pathogenesis with multiple and chemically diverse lipid species (sphingolipids, ceramides, sphingomyelins, gangliosides, trans fatty acids, etc.). In general, APP can be processed differently depending on the physical state of the surrounding membrane and composition of lipid species [2,3]. Interaction of Aβ peptides with cellular membrane mimetics was shown to be regulated by phase transitions between liquid-ordered and liquid-disordered or liquid-expanded and liquid-condensed states, with the considerably more effective binding to the liquid-disordered phase further facilitated by the presence of negatively charged lipids [4,5].

Cholesterol differs from any other lipid present in cellular membranes with respect to many aspects. Due to its unique molecular geometry, it regulates the formation of ordered lipid clusters (also known as rafts) and through specific interactions with the intramembrane regions of various proteins controls the affinity of the proteins to the clusters [6]. Cholesterol interactions with APP itself and virtually all its cleavage products are arguably the most complex and functionally versatile, as evidenced by a large proportion of publications emphasizing their role in different AD-related processes. For example, genetic variation in a cholesterol transport protein, apolipoprotein E (apoE), is the most common genetic risk factor for sporadic AD. Moreover, the knockdown of cholesterol synthesis in astrocytes was found to significantly shift the balance between the cleavage of APP by secretases towards α-secretase, presumably by driving APP out of the ordered lipid clusters (rafts) where functional β- and γ-secretases reside [7]. Secretase enzymes, cleavage by which is a source of C99 and a broad spectrum of other peptides, were shown to be sensitive to the local cholesterol content in the membrane [8]. Amyloid-β (Aβ) peptides, the ones probably most frequently discussed in the context of AD pathogenesis, are also known to directly interact with cholesterol molecules, and such interactions play a notable role in plasma membrane pore formation, among other processes that involve the participation of Aβ [9]. Another cleavage product attracting a lot of attention, the APP transmembrane (TM) fragment C99, was also shown to interact (relatively weakly) with cholesterol, apparently competing for the interaction site with the C99 dimerization [10]. Several cholesterol-binding sites on the surface of C99 and its fragments have been described and are situated around the glycine zipper motif in the N-terminal part of the transmembrane (TM) helix and the loop connecting it to the flexible juxtamembrane (JM) helical region or in the C-terminal flexible cytoplasmic domain [9,11,12,13,14]. However, since APP processing by different secretases tends to occur in diverse membrane environments distinguished by lipid composition and ordering state, it is natural to assume that the interactions with cholesterol, as well as their functional implications, also depend on the properties of the local lipid environment. Indeed, in [15,16], cholesterol concentration and lipid phase were found to affect the conformational ensemble of C99 and its fragments, although the authors failed to find any specific binding modes for cholesterol. In addition to the “normal” orientation in the membrane roughly perpendicular to its surface plane [17,18], cholesterol is known to be capable of adopting various “tilted” orientations [19,20], including the orientation parallel to the surface originally observed in the polyunsaturated lipid bilayers having more degrees of lipid tail freedom [21,22]. In this study, using high-resolution NMR spectroscopy, we found that a spin-labeled cholesterol analog, 3-β-doxyl-5-α-cholestane (hereinafter doxyl-cholestane), is able to interact with APP roughly in the middle of its TM helix in the immediate vicinity of the flexible hinge elongated from the helix-destabilizing diglycine insert G^708^G^709^ to approximately A^713^ of the helix [13,23,24,25]. This interaction is enhanced in the more disordered lipid environments, represented in our experiment by detergent micelles, whereas in the more ordered lipid bicelles the “classical” cholesterol interaction sites are predominant. It appears that, being in the proximity of the new “central” interaction site, the sterol molecule can adopt the same orientation parallel to the membrane plane as cholesterol in the polyunsaturated membranes. Molecular dynamics (MD) simulations suggest the ability of the sterol molecule to travel between the three alternative interaction sites situated in the same weakly polar groove on the TM helix surface, and the interaction through each of the three sites is accompanied by subtle structural adjustments that can play a role in the processes of APP cleavage by secretases and thus be directly relevant to the AD initiation. Moreover, the comparison of the results obtained in micellar and bicellar environments revealed the modulation of the specific interaction of APP with individual lipids, such as sterols, by the overall properties of the lipid environment, implying that the lipid composition of plasma membranes and local perturbations of the membrane properties of any genesis can be of consequence for AD progression.

## 2. Results

### 2.1. Sterol Interaction with APP TM Domain in Membrane-Mimetic Environments by NMR

In order to evaluate the spectrum of possible interactions with sterol molecules, we investigated a wild-type recombinant APPjmtm peptide corresponding to the APP_686–726_ fragment including the intact TM domain with the adjacent N-terminal JM region. This was performed using heteronuclear NMR spectroscopy in two different membrane-mimetic environments (detergent micelles and lipid bicelles) in the presence of a spin-labeled analog of cholesterol, 3-β-doxyl-5-α-cholestane. The structural formula of 3-β-doxyl-5-α-cholestane (doxyl-cholestane) is shown in Figure 1.

To obtain the baseline NMR spectra for comparison, in the control experiments, the intensity of the nitroxide spin-label was quenched by ascorbic acid. Panels (A) and (B) of Figure 1 illustrate the inhomogeneous broadening of the heteronuclear NMR ^1^H/^15^N-TROSY signals caused by interactions with the unquenched labels in micellar and bicellar environments, respectively (the signals with quenched and unquenched labels are overlaid in each of the panels). The distribution of the signal broadening caused by proximity of the doxyl group along the amino acid residue sequence of the peptide is shown in the form of a bar chart on panels (C) and (D) of Figure 1 for micelles and bicelles, respectively. Clearly, the signals from the residues in the JM region (V^689^–A^692^) are strongly affected by the proximity of the doxyl group in both membrane mimetics. The G^696^ and N^698^ residues in the loop region are also affected by the interaction with doxyl group in both cases, with the effect being somewhat stronger in micelles.

The interaction patterns of the residues inside the membrane with the sterol analog, however, are notably different between the cases of bicelles and micelles. In bicelles, the maximal broadening (associated with the maximal proximity of the spin-labeled headgroup) tends towards the N-terminal part of the G^700^xxxG^704^xxxG^708^ glycine zipper motif composed of the residues with small side-chains (as a part of the so-called tandem GG4-like motif [26]), which is consistent with the sterol alignment along the helix axis, roughly perpendicular to the membrane surface. In micelles, there are four distinct minima on the bar chart, associated with the maxima of proximity of the corresponding residue amide groups to the sterol headgroup. The spacing between them corresponds to one helix turn, and the overall pattern implies that there are several sterol interaction sites, each of them contributing to two or three neighboring minima so that the resultant picture is a weighted average of contributions from all the binding sites. The weighing factors are the functions of occupancies of each respective site. At the very least, the observed pattern implies the appearance of another sterol headgroup interaction site including residues G^709^xxxA^713^ (composed of the central GG4-like motif of the APP TM domain) in the micellar membrane-mimetic environment, which corresponds to a more disordered state of the lipid tail groups and water permeability.

Based on the marginal broadening of the signal on the C-terminal side of the peptide, it was concluded that the sterol weakly interacts with the TM segment via its headgroup oriented towards the end of the helix. It seems unlikely that the difference in the side-chain sizes between different residues distorts the spatial pattern of the proximity of the spin-label group to the backbone amide groups; since according to [27], signal broadening beyond the resolution occurs at the distances of up to 14 Å (assuming the permanent proximity of the label).

### 2.2. Sterol Interaction with APP TM Domain in Explicit Lipid Bilayer by MD

In order to evaluate the ability of a sterol molecule to interact with different putative sites on the surface of the APPjmtm peptide, a series of independent unconstrained MD simulations was performed in the explicit POPC bilayer containing one molecule of doxyl-cholestane, which was initially bound and relaxed with the geometric constraints corresponding to the three putative binding sites suggested by the NMR data (Figure 2, Appendix A).

As can be seen on Figure 2A,C, in the unconstrained simulations, the sterol transiently interacted with the peptide with relatively small, but statistically significant lifetimes. In all the cases, the sites of APPjmtm interaction with the sterol headgroup were clustered around three locations with a satisfying accuracy corresponding to the interaction sites suggested by the NMR data (Figure 2B), namely, the site under the JM helix, the N-terminal G^700^xxxG^704^ motif of the TM helix, and the central part of the TM helix near the G^709^xxxA^713^ motif (Figure 2D). Furthermore, an unconstrained 1000 ns MD simulation of the APPjmtm peptide with 20 doxyl-cholestane molecules freely embedded in the POPC bilayer also revealed multiple transient sterol–protein interactions (up to hundreds of nanoseconds long) at similar sites identified by NMR (Figure 3). Additionally, the transient sterol–protein interactions occurred near the C-terminal part of the APP TM domain where the marginal signal broadening (more pronounced in the bicellar environment) caused by the proximity of doxyl-cholestane was detected in the NMR spectra.

The secondary structure evolutions during the constrained and unconstrained relaxations (Figure 3 and Figure 4, Appendix A) are indicative of a potentially significant pattern of behavior of certain peptide moieties capable of structural rearrangements. As can be seen in Figure 4A, the presence of the sterol molecule in the vicinity of the JM region stabilized its α-helical structure. Constrained MD relaxation with the sterol near the hinge region revealed a tendency of the sterol to stabilize the kinked configuration of the TM helix (Figure 4B), which was preserved (with some variations) throughout the entire simulation (unlike the rest of the MD relaxation traces). As soon as the constraints were relieved, the sterol molecule rapidly (after ~3 ns) moved into the bulk of the lipid bilayer. Such an escape almost precisely coincided with restoration of the α-helical structure (after ~7 ns) in the hinge region, and was closely followed by a slight (1–2 residues) shortening of the α-helix on the C-terminal side (Figure 4B). The constrained configuration apparently imposed some stress on the system and judged the relatively rapid straightening of the helix and the removal of the sterol from the interaction with the site. However, since the sterol molecule was observed to freely migrate into the vicinity of the same interaction site during MD simulations starting from an entirely different system configuration (Figure 2), this interaction site cannot be considered as a modeling artifact. Consistency of the NMR signal broadening with the observed pattern is another argument in favor of the existence of such an interaction mode. In all the simulations, the JM region was almost continuously in contact with the membrane lipids; however, the secondary structure of the region and its spatial orientation with respect to the membrane surface were slightly but detectably dependent on the sterol interaction with the protein, as especially and lucidly evidenced by MD relaxations with constraints imposed on the location of the sterol molecule in the vicinity of each interaction site.

## 3. Discussion

The experimental part of the present study was performed using a spin-labeled derivative of cholesterol. The obvious advantage of this choice is the ability to attribute with high credibility certain observed changes in the NMR signals due to the proximity of the sterol molecule polar headgroup to each of the amino acid residues. This provides additional information about the orientation of cholesterol relative to the APP TM domain axis. The equally obvious trade-off is the potentially disputable applicability of the results to the evaluation of the peptide interaction with the unmodified cholesterol in vivo. For example, the doxyl-cholestane molecule tends to be localized somewhat deeper in the membrane compared to the unmodified cholesterol, reflecting the less polar properties of the doxyl group [28]. However, as the doxyl-cholestane interaction sites of the APP TM domain coincide with the native cholesterol interaction sites [9,10,11,12,13], it is reasonable to assume that the essential traits of cholesterol interactions are faithfully represented by the spin-labeled analog. In addition to that, two distinct groups of oxysterols are known to be formed under oxidative stress conditions, i.e., ring-oxidized sterols (mostly produced by free radicals) and tail-oxidized sterols (mostly produced enzymatically), and to behave differently in a lipid membrane. Unlike tail-oxidized sterols or cholesterol, ring-oxidized sterols can efficiently acquire tilted orientations in the bilayer leading to a stronger disruption of the membrane structure [29]. In addition, in the polyunsaturated lipid bilayer environment, cholesterol was shown capable of assuming tilted orientations [19,20] similar to that observed for doxyl-cholestane interacting with the novel “mid-membrane” interaction site on the surface of the APP TM domain. In other words, the lipid composition (e.g., the content of sphingomyelin, ganglioside, polyunsaturated lipid species, and other components altering the membrane phase state) or protein–lipid interactions affecting local membrane properties (phase state, curvature, etc.) can alter the modes of interaction of cholesterol with APP. The significance of the new cholesterol interaction site can vary with the cholesterol content differing greatly between the plasma membrane, where most of cholesterol resides, and ER membranes where the cholesterol-regulating machinery operates [30]. Thus, even the properties of doxyl-cholestane that differ from those of native cholesterol appear to be biologically relevant under certain conditions. Moreover, the concerns about the applicability are greatly alleviated by the consistency between our data for doxyl-cholestane and the published results of the NMR experiments and MD simulations performed by other groups with cholesterol and its analogs, wherever there is a basis for a meaningful comparison. In particular, the experimental data published in the recent studies [9,11,12,13] with cholesterol and its micelle-forming analogs corroborate the existence of the interaction sites situated near the G^700^xxxG^704^xxxG^708^ glycine zipper motif of the APP TM domain and near the adjacent extracellular loop region, corresponding to our site 2 and site 1, respectively. Although the sterol headgroup interaction with the site including G^709^xxxA^713^ residues (the novel site 3) was not explicitly discussed in any of these publications, chemical shifts were additionally observed around the diglycine insert G^708^G^709^ in the study [13], where bicelles containing up to 20% cholesterol were used as a membrane mimetic. In the absence of a spin-label in the sterol headgroup, such a change in the backbone amide group, chemical shifts cannot be with any degree of credibility associated with a separate cholesterol-binding site. It can be equally attributed to the local structural rearrangements of the protein itself (e.g., the central kink straightening) that are induced by the cholesterol interaction via the N-terminal sites.

Our experimental NMR data, however, strongly suggest that sterols can interact directly with this site through their headgroups. Figure 5A,C represent the NMR signal broadening induced by the spin-label mapped onto the APPjmtm surface for micelles and bicelles, respectively. The sterol–protein interaction patterns for the cases of micelles and bicelles share certain common features, including one almost perfectly identical interaction site on the N-terminal side, but the micelles are distinguished by an additional site in the center (the hinge region) and a slight difference in the extent of interaction in the JM region. As illustrated by Figure 5B, all the interactions occur around the small side-chain residues, exposing the polar backbone and forming on the peptide surface a continuous relatively polar groove (Figure 5D,F) structurally composed of a condensed system of GG4-like motifs with a single diglycine insert G^708^G^709^. These panels of Figure 5 show the 3D surface of the protein color-coded in accordance with the molecular hydrophobicity potential (MHP) distribution [31], providing a clue as to how the fairly polar headgroup of a sterol molecule can freely sample the hydrophobic intramembrane core to access the deeply buried binding sites. The extended relatively polar glycine zipper groove formed by the backbone atoms of glycine residues can be exploited by the sterol to slide along the TM helix.

Based on the difference in the patterns of signal broadening induced by spin-label interactions in the micelles and bicelles, the less ordered aliphatic chain packing makes the sterol presentation in the center of the membrane hydrophobic core more likely. Indeed, in the experiments with polyunsaturated lipid bilayers published in [21,22], cholesterol was demonstrated to be able to lie flat in the middle of the membrane, as opposed to the “conventional” upright orientation common for more ordered lipid bilayers (e.g., highly enriched with cholesterol). The upright orientation was reported to be generally preserved upon the replacement of the hydroxyl headgroup by the doxyl moiety as in doxyl-cholestane. Furthermore, although doxyl-cholestane has a preferred membrane orientation, it can also rotate about its short axis, transitioning between the “upside down” and the “upright” orientations. The differences between micelles and bicelles as physical models of biological membranes can reflect the different local states of the lipid bilayer, with the micelles corresponding to the more disordered and water-permeable spots, which can be induced, for example, by protein–lipid interactions perturbing the membrane or occur at the boundaries of the membrane microdomains. Switching between the bicellar and micellar membrane-mimetic environments was reported to cause a transition between the two alternative biologically relevant conformations (corresponding to the active and inactive receptor states) of the TM domain of type I receptor EGFR [32,33]. The observed sterol interaction sites overlap with the alternative dimerization interfaces of the APP TM domain (which is also often attributed to type I receptors [34]). The micellar environment strongly favors dimerization via the central G^709^xxxA^713^ motif [23], whereas in the liposomes [35,36], the N-terminal G^700^xxxG^704^xxxG^708^ glycine zipper motif is apparently employed (Figure 5D–F). Clearly, there is some competition between the relatively weak dimerization of the APP TM domain [10,23] and its interaction with sterols, including cholesterol. A little less obvious, but arguably as important, subtle changes in the lipid environment can modulate both the preferred dimerization mode and cholesterol accessibility, thus creating a complex system of feedbacks. One such feedback mode is suggested by the recently proposed novel role of APP C-terminal fragment presumably acting in the endoplasmic reticulum as a cholesterol-sensing peptide, promoting the formation of local ordered domains [30]. The conformation of this cytoplasmic juxtamembrane region can be altered by the APP TM dimerization mode, thus creating a feedback loop between cholesterol concentration, its local distribution, and concentrations of its esterified forms. According to MD simulations, the lipid composition of the bilayer modulates the relative stability of the competing dimeric configurations [37,38].
Figure 5The degree of surface hydrophobicity of the APPjmtm peptide and the surface distribution of its interaction with the sterol headgroup in different membrane mimetics. (**A**,**C**) Amide signal broadening induced by the spin-label is mapped onto the APPjmtm surface for micelles and bicelles, respectively, using the color coding shown by the scale bar on the left side (the surface is colored in gray where the information about signal broadening is not unambiguously attributable). The green and yellow colors of the schematic representation of membrane planes reflect a certain difference in the properties of the membrane mimetics (more polar micelles vs. more ordered bicelles having a small spot of “quasi-bilayer”). (**B**) The ribbon representation of the APPjmtm surface viewed from the same points as in (**A**,**C**), with glycine and alanine residue locations highlighted in green. (**D**,**F**) Ther surface properties of the alternative APP TM domain dimer interfaces reported for micellar (PDB ID: 2loh) [23] and liposomal [35,36] environments. The hydrophilic (polar) and hydrophobic surfaces of the APPjmtm are colored in green and yellow, respectively, according to their surface molecular hydrophobicity potential (MHP) values [31] as shown on the scale bar on the left. The complementary subunits of the dimers are shown in a stick representation. (**E**) The hydrophobicity map of the molecular surface of the APP TM helix with the isolines encircling hydrophobic regions with high MHP values. The map is presented in cylindrical coordinates associated with the TM helix (as described in [23]). The alternative N-terminal (more polar) and central (more hydrophobic) helix packing interfaces of the APP TM domain are encircled by green and orange ovals, respectively. Residues composing characteristic dimerization GG4-like motifs G^700^xxxG^704^xxxG^708^ and G^709^xxxA^713^ are highlighted in green and orange, respectively.
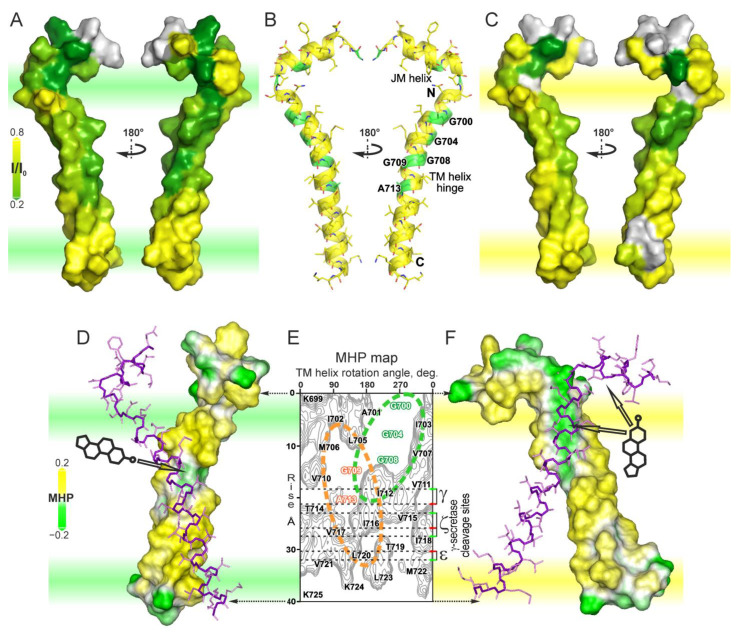



Thus, cholesterol enrichment can not only cause the alterations in the APP TM domain conformation with a potential pathophysiological significance, but also the exact composition and physico-chemical state of the remaining lipids [39,40] can either potentiate or mitigate the effects of cholesterol on the conformation of the TM domain (Figure 6). Moreover, the effects of cholesterol interactions can have consequences not only for the secretase cleavage cascades (Figure 1 and Figure 5E) but also for the functioning of the full-length APP in the plasma membrane.

According to our NMR and MD simulation data (Figure 2 and Figure 3), the sterol molecule sequestered in the vicinity of the hinge region can assume the orientation parallel to the plane of the membrane and possibly stabilize a strongly bent configuration of the hinge (Figure 4B), depending on the local state of the lipid bilayer. Thus, there appears to be a certain correlation between the perturbed state of the membrane and the “horizontal” presentation of sterols in the membrane midplane (as shown in Figure 6A). The stabilization of the kinked configuration of the APP TM domain in the locally perturbed or polyunsaturated lipid-enriched environment allowing the horizontal presentation of sterols has to be accompanied by the elongation of the α-helical structure on the C-terminal end. On the other hand, the stabilization of the straightened hinge configuration in a more ordered lipid environment (favoring “vertical” sterol presentation) by the interaction of a sterol (cholesterol in vivo) with the N-terminal part of the APP TM helix can cause the local unfolding of the α-helical structure on the C-terminal end near the ε-cleavage site, thereby exposing a different peptide bond for cleavage by γ-secretase, which is known to occur around the lipid polar head plane (Figure 6B). This can shift the balance between the production of different Aβ specimens in favor of the Aβ42 form. This effect is dependent on the cholesterol concentration and the presence of its oxidized forms, with the high levels of cholesterol favoring its “vertical” orientation corresponding to the described interaction modes and esterification enabling the “horizontal” presentation of the sterol towards the central part of the membrane. It has been reported that a cholesterol content of more than 15% in the lipid raft pool is sufficient to disrupt C99 dimerization and trigger the monomer-dependent γ-processing [41].

In addition, sterol interaction via the N-terminal sites appears to enhance the helicity of the amphiphilic JM region lying on the membrane surface. Indeed, as reported in [36], cholesterol and “Flemish” A692G mutation can act synergistically, increasing both the JM region helicity and Aβ peptide secretion. Hence, in the cholesterol-enriched microdomains of the plasma membrane, the N-termini of the JM helix would be protected from cleavage by α-secretase, whereas the accessibility of the β-secretase cleavage site would remain unchanged, forcing the cleavage cascade to shift towards the Aβ production. Indeed, the α- and β-secretases are known to have opposite functional preferences between the non-raft and raft regions of the membrane [1]. Generally speaking, the accessibility of APP for α-, β- and γ-secretase recognition and cleavage significantly depends upon local membrane properties including the content of cholesterol and its esterified forms that would interact differently with monomeric APP and its different dimers having varied exposure of the observed cholesterol/sterol-binding sites. Although the two previously known interaction sites in the APP N-terminal part of the TM domain and adjacent amyloidogenic JM region remain viable targets for AD drug development, the information about the novel binding site for sterol molecules, occupancy of which would modulate APP processing by secretases, suggests a possible strategy for development of sterol-based AD therapeutic agents targeting the novel site to avoid the pathological processing of APP.

## 4. Materials and Methods

### 4.1. NMR Spectroscopy in Micellar Environment

The ^15^N/^13^C-labeled recombinant peptide APPjmtm (*M*Q^686^KLVFFAEDVGSNKG^700^AIIGLMVGGVVIATVIVITLVML^723^KKK^726^, APP_686–726_ fragment, the hydrophobic TM segment) was obtained according to [42] and solubilized in an aqueous suspension of dodecylphosphocholine (DPC; Avanti Polar Lipids (Alabaster, AL, USA)) micelles or dimyristoylphosphocholine/dihexanoylphosphocholine (DMPC/DHPC) bicelles (Avanti Polar Lipids (Alabaster, AL, USA)) at monomeric conditions with the lipid/peptide of ~160 [23]. The effective molar ratio q of long- and short-chain lipids in the DMPC/DHPC bicelle was ~0.3, assuming a free DHPC concentration of 7 mM in the bicellar suspension. The peptide powder was first dissolved in 2:1 (*v*/*v*) trifluoroethanol–water mixture with the addition of DPC or DMPC/DHPC, placed for ten minutes in an ultrasonic bath, and then lyophilized. After that, the dried APPjmtm samples were dissolved at pH 5.7 in a buffer solution containing 15 mM citric acid, 40 mM Na_2_HPO_4_, and 5% D_2_O (*v*/*v*). The spin-labeled analog of cholesterol, 3-β-doxyl-5-α-cholestane (Sigma-Aldrich, St. Louis, MI, USA), was initially dissolved in chloroform and mixed with the chloroform solutions used for preparation of the DPC micelles or DMPC/DHPC bicelles (q ≈ 0.3). After the evaporation of chloroform, micelles or bicelles containing doxyl-cholestane were formed and added to the 0.2 mM APPjmtm samples to the final lipid/sterol ratio of ~80 that corresponds approximately to one spin-labeled sterol molecule embedded into each membrane-mimicking particle. In order to ensure the uniformity of distribution of APPjmtm and sterol molecules in micelles and bicelles, several freeze–thaw cycles were carried out, followed by sonication. High-resolution heteronuclear NMR spectra of the 0.2 mM APPjmtm samples were acquired at 318 K 800 MHz AVANCE III spectrometers (Bruker BioSpin, Rheinstetten, Germany) equipped with pulsed-field gradient triple-resonance cryoprobes. The spectra were analyzed with the CARA software (http://cara.nmr-software.org, accessed on 30 December 2021) [43]. The backbone ^1^H, ^13^Cα, and ^15^N resonances of APPjmtm at pH 5.7 in the micelles and bicelles were assigned using the BEST-TROSY version [44] of triple-resonance ^1^H/^13^C/^15^N-HNCA and ^1^H/^13^C/^15^N-HN(CO)CA experiments based on the APPjmtm chemical shifts previously obtained at pH 6.2 in the DPC micellar environment [23]. Spectra were recorded with the non-uniform sampling of indirect dimensions and processed using the qMDD software (https://github.com/TheFausap/QMDD, accessed on 15 January 2016) [45]. In order to characterize the interaction of the sterol molecule with the APPjmtm peptide, the relative decrease in the ^1^H/^15^N-cross-peak intensity due to signal broadening caused by the proximity of the doxyl group along the amino acid residue sequence of the peptide was analyzed in the ^1^H/^15^N-TROSY spectra. To obtain the baseline NMR spectra for comparison, in the control experiments, the intensity of the nitroxide spin-label was quenched by 2 mM ascorbic acid. 

### 4.2. Molecular Dynamics in Explicit Lipid Bilayer

In order to elucidate the details of the NMR-observed interactions of the APP_686–726_ fragment with the spin-labeled analog of cholesterol, 3-β-doxyl-5-α-cholestane, molecular dynamics (MD) simulations were performed in an explicit lipid bilayer using the GROMACS 5.1.4 package [46] and Amber99SB-ILDN force field [47] with the TIP3P water model [48] and lipid parameters as described elsewhere [47]. The initial monomeric conformations were derived from the NMR structure of the APPjmtm peptide embedded into DPC micelles (PDB ID: 2llm [11]). The unprotonated states of ionogenic side-chains of the E^793^ and D^794^ residues were used corresponding to pH ~5. The starting configurations of the simulated systems were obtained by inserting the APP TM fragment into a pre-equilibrated lipid bilayer consisting of 200 1-palmitoyl-2-oleoyl-sn-glycero-3-phosphocholine (POPC) and 1 3-β-doxyl-5-α-cholestane molecules. The systems were solvated, and counterions were added to make them electrically neutral. The molecular topology of the doxyl-cholestane was created based on the SLIPIDs force field [47]. Partial charges for the atoms of the doxyl part of cholestane were chosen by analogy with standard phospholipids [49]. If two different atoms form a covalent bond, the partial charge σ shifts toward the less electronegative atom. The σ values for the couples NO, CO, CH, and OH were 0.1, 0.2, 0.1, and 0.45 electron charges, respectively, as described in [50]. The alternative initial configurations of the protein–membrane system for each MD relaxation were selected based on the analysis of NMR signal broadening caused by the presence of doxyl-cholestane in micelles and bicelles.

The positions of the doxyl-cholestane molecules were initially constrained using the linear/harmonic approach with the force constant of 1000 kJ·mol^−1^·nm^−2^, within the distance interval of 8–10 Å (distance between the sterol doxyl group, O^N^ atom, and the protein amide groups, H^N^ atoms) from three groups of amino acid residues V^689^/F^690^/F^691^, G^696^/A^701^/G^704^, and L^705^/G^709^/A^713^ corresponding to the three characteristic patterns of signal broadening observed in the NMR spectra with micelles and bicelles. The patterns corresponded to the center of the JM helix, the N-terminal G^700^xxxG^704^ motif of the TM helix, and the central part of the TM helix near the G^709^xxxA^713^ motif. After 20 ns constrained MD relaxation, the constraints were removed, and MD calculations were continued during 100 ns for all the systems with a 2 fs integration time step. The procedure was repeated four times for each characteristic pattern. In addition, two unconstrained 1000 ns MD simulations were carried out for the APP TM fragment embedded into the pre-equilibrated POPC bilayer containing initially free 20 doxyl-cholestane molecules. The spherical cut-off function (12 Å) was used for the truncation of van der Waals interactions, while the electrostatic interactions were treated using the particle-mesh Ewald summation (real space cut-off 12 Å and 1.2 Å grid with the fourth-order spline interpolation). MD simulations were carried out with the imposed 3D periodic boundary conditions in the isothermal–isobaric (NPT) ensemble with the semi-isotropic pressure of 1.013 bar scaled independently along the bilayer normal and in the bilayer plane at the constant temperature of 310 K. Temperature and pressure were controlled using Nose–Hoover thermostat and Parrinello–Rahman barostat with 0.5 and 10 ps relaxation parameters, respectively, and the compressibility of 4.5 × 10^−5^ bar^−1^ for the barostat. The temperatures of the protein, lipids, and water molecules were coupled separately. The conformational dynamics of the protein and its van der Waals contacts with lipid and water molecules were analyzed using the GROMACS package utilities. In order to map the sterol–protein interactions, the numbers of direct contacts between heavy atoms within 4 Å distance cut-off were estimated. The NMR and MD simulation data were analyzed and visualized with MOLMOL [51] and PYMOL (Schrödinger, LLC (New York, NY, USA)).

## 5. Conclusions

In summary, our results appear to suggest that the spectrum of possible interactions of APP and its cleavage products with sterol molecules, including cholesterol, is somewhat broader than can be deduced from the previously published structural and modeling studies. The interactions can be essential both for the choice of secretase cleavage-processing pathways crucial for Alzheimer’s disease pathogenesis and for the ostensibly multifaceted (and not yet fully explored) normal function of APP. Moreover, the protein transmembrane domain interactions with cholesterol appear to be modulated by the local physico-chemical state of the lipid environment (liquid-ordered, liquid-disordered, phase-separation boundary) and the chemical composition of the surrounding lipids (sphingolipids, ceramides, sphingomyelins, gangliosides, trans fatty acids, etc.). This suggests that inclusion of a proper combination of minor lipid species in the diet, and the regulation of brain cholesterol levels can provide a substantial boost to the existing strategies of reactive remediation and proactive prevention of AD and other age-related human pathologies. It can apply equally to pharmaceutical treatment and dietary supplementation.

## Figures and Tables

**Figure 1 ijms-26-00553-f001:**
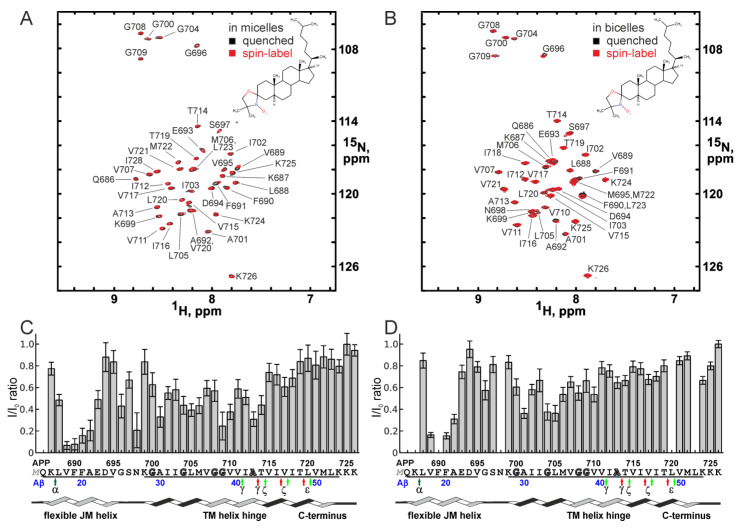
The NMR spectra of the APPjmtm peptide in DPC micelles and DMPC/DHPC bicelles. (**A**,**B**) The two-dimensional 1H/15N-TROSY NMR spectra of the APPjmtm peptide embedded in DPC micelles and DMPC/DHPC bicelles, respectively, in the presence of doxyl-cholestane are shown in red. The reference spectra obtained after quenching the spin-label (doxyl group) with the aid of ascorbic acid are shown in black. The structural formula of doxyl-cholestane is presented in the top right corner. (**C**,**D**) The signal broadening of APPjmtm amide groups caused by the spin-label quantified as the ratio of the normalized unquenched and quenched signal intensities for DPC micelles and DMPC/DHPC bicelles, respectively. Errors bars reflect the noise level of the measurements. Dual amino acid residue numbering is provided, corresponding to the full-length APP (black) and Aβ peptides (blue). The alternative γ-secretase cleavage cascade sites [24] leading to the stepwise production of Aβ42 and Aβ40 peptides are shown by red and green arrows, respectively. The dark green arrow indicates the α-secretase cleavage site. The schematics of the secondary structure deduced from our own studies [11,23] and experimental data published by other groups [13,25] are provided at the bottom. The helical region with enhanced internal dynamics is highlighted in gray. The residues corresponding to the characteristic dimerization of GG4-like motifs G700xxxG704xxxG708 and G709xxxA713 are marked in the figure. Schemes follow the same formatting.

**Figure 2 ijms-26-00553-f002:**
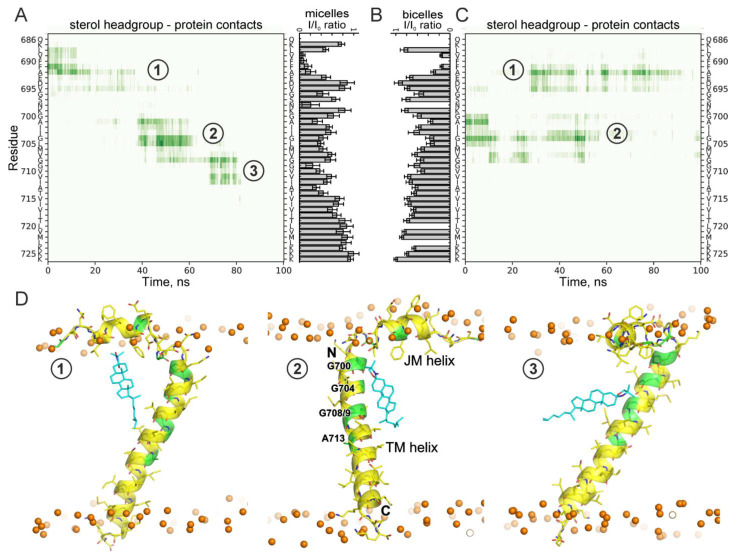
The sites of sterol interaction with the APPjmtm peptide. (**A**,**C**) The representative maps of amino acid residue contacts with the sterol polar headgroup derived from 100 ns unconstrained MD simulations in the explicit POPC bilayer started after 20 ns constrained simulations, during which the sterol molecule was restrained near the JM helix (**A**) and near the N-terminal G^700^xxxG^704^ motif (**C**) (see also Appendix A). The intermolecular contacts are color-coded according to the number of direct van der Waals contacts between atoms within 4 Å distance: from white (0 contacts) to green (20 contacts). During the simulations, the sterol molecule migrated freely between three different interaction sites, two of which (near the JM helix (1) and near the N-terminal motif G^700^xxxG^704^ (2)) were described earlier [9,11,12,13] and were observed in our experiments with bicelles, and one more (3) corresponded to the additional signal-broadening pattern observed in the micellar environment near the central motif G^709^xxxA^713^. The bar charts of amide signal broadening due to the interaction with spin-label measured in micelles and bicelles are presented in panel (**B**) for comparison. (**D**) Bottom panels are the corresponding representative MD snapshots of the peptides in the POPC bilayer (only the phosphorus atoms of the lipid headgroups are shown by orange spheres). The peptides are in the ribbon presentation, glycine and alanine residues are shown in green, and the rest of the sequence is in yellow. The sterol molecule is shown in cyan. The residues corresponding to characteristic dimerization GG4-like motifs G^700^xxxG^704^GxxxG^708^ and G^709^xxxA^713^ are indicated in the central snapshot.

**Figure 3 ijms-26-00553-f003:**
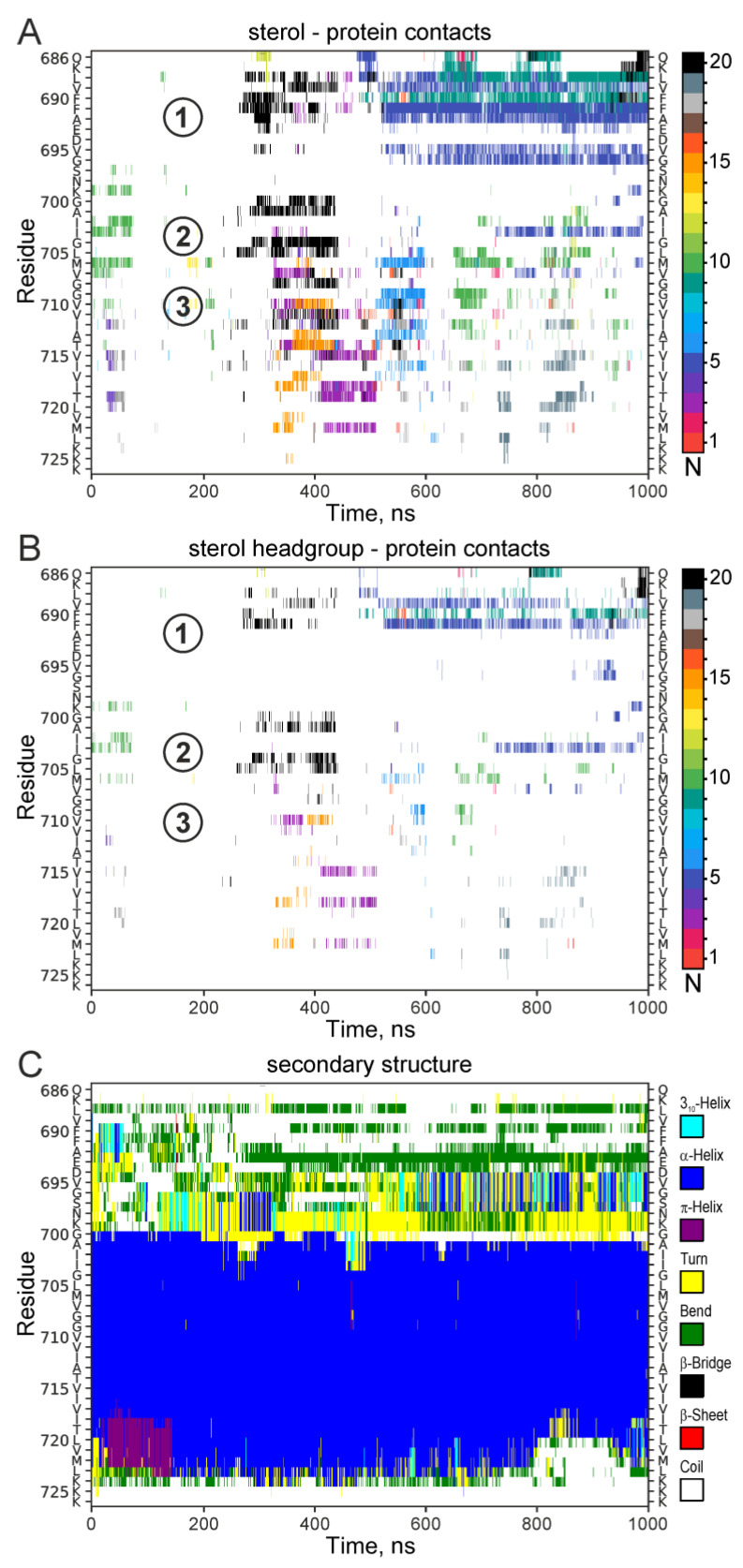
The multiple interactions of sterols with the APP TM domain. Evolutions of direct contacts of the APPjmtm peptide with 20 doxyl-cholestane molecules (with 4 Å distance cut-off between all heavy atoms (**A**) or with the polar headgroup only (**B**) and the protein secondary structure (**C**)) during unconstrained 1000 ns MD simulations in the POPC bilayer. The contacts from different sterol molecules (numbered as N) and the secondary structure elements are color-coded consequently as shown by the vertical bars on the right. The positions of sterol-binding sites 1–3 identified by NMR are marked.

**Figure 4 ijms-26-00553-f004:**
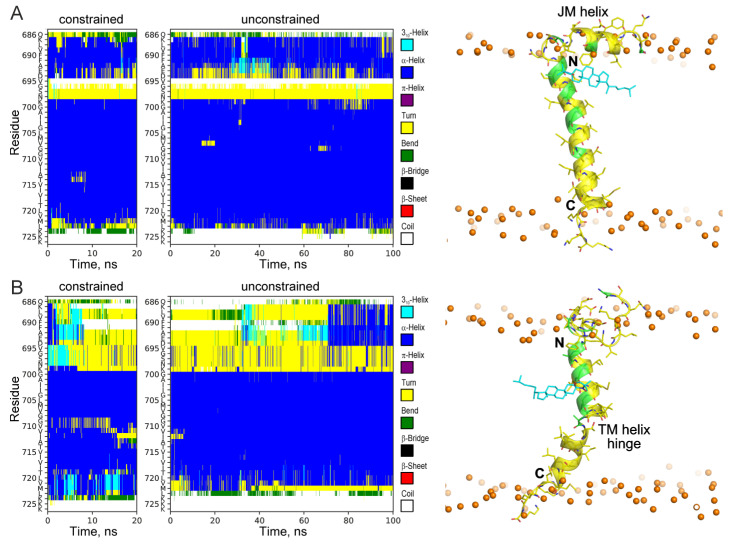
The secondary structure evolutions of the APPjmtm peptide during constrained and unconstrained MD relaxations. The color-coded representation of time evolution of the peptide secondary structure is presented. (**A**) The sterol molecule was initially constrained (for 20 ns) near the N-terminal G^700^xxxG^704^ motif and then released for 100 ns MD relaxation. A representative snapshot of the peptide in the POPC bilayer is shown on the right, and the notation conventions are as in Figure 2. The sterol molecule interacted with the polar headgroup with the N-terminal G^700^xxxG^704^ motif packed into a hydrophobic pocket under the folded JM helix (indicated on the snapshot), which was described previously in [11]. (**B**) The sterol molecule was initially constrained (for 20 ns) near the G^709^xxxA^713^ motif in the central part of the TM helix and then released for 100 ns MD relaxation. During the constrained simulation and the first ~7 ns, the configuration illustrated by the snapshot on the right was maintained. The hinge position (marked on the snapshot) coincided with the position previously observed by solid-state NMR in liposomes [25]. The sterol molecule occasionally adopted the orientation shown in the snapshot (parallel to the bilayer plane).

**Figure 6 ijms-26-00553-f006:**
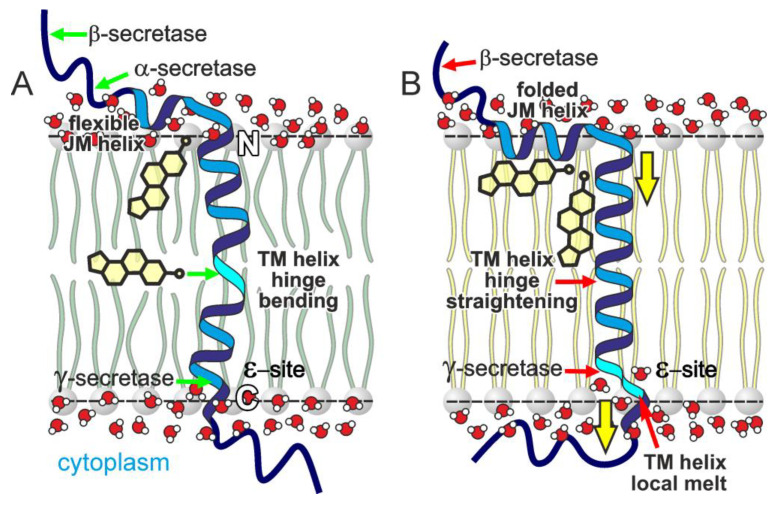
A model illustrating the relations between the sterol interaction and structure-dynamic properties of the APP JM-TM segment in different states of the local lipid environment. (**A**) The bending of the central hinge region (in cyan) favors “normal” APP cleavage cascades (enhancing the Aβ40 production; see text for explanation). A more disordered state of the membrane (the orientation of lipid tails and water permeability) facilitates access of sterol to the flexible hinge region, horizontal sterol presentation, and certain elongation of the TM helix on the C-terminal side due to the kinked structure of the helix. (**B**) The straightening of the central hinge region favors “abnormal” APP cleavage cascades. The raft-like, more ordered state of the membrane forces cholesterol in the hydrophobic core to be oriented along the tightly packed lipid tails, creating a bias towards the straightened configuration of the hinge region accompanied by C-terminus unfolding and alternative exposure for γ-secretase cleavage. At the same time, the interaction with the nascent JM helix can stabilize its fold, changing its accessibility for α-secretase cleavage. Thus, the reallocation of cholesterol inside the plasma membrane self-consistent system can serve as a trigger of different cleavage-processing pathways of APP that delay or accelerate the AD onset. The yellow arrows show the direction of presumable displacement of the polypeptide chain relative to the surfaces of membrane leaflets, green arrows designate “normal” processes, and red arrows—their pathological “abnormal” variants.

## Data Availability

Data are contained within the article or Appendix A.

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
