# Peer review of "Diverse Interactions of Sterols with Amyloid Precursor Protein Transmembrane Domain Can Shift Distribution Between Alternative Amyloid-β Production Cascades in Manner Dependent on Local Lipid Environment"

_ijms, 2025, doi:10.3390/ijms26020553_

Round 1

Reviewer 1 Report

Comments and Suggestions for Authors

The manuscript is well written and the study design is well structured and organized. The purpose of the study is very interesting and could contribute to the current knowledge about the pathways involving APP.

In the present manuscript, the Authors evaluated the critical link between Alzheimer’s disease (AD) pathology and lipid metabolism, considering the contribution of cholesterol in processing of Amyloid Precursor Protein (APP) and introducing for the first time the critical role of the entire lipid subsystem rather than the effects of individual lipid species.

The Introduction well summarized the current knowledge about the role of lipid metabolism and AD etiopathology, and described the aims of the work, giving to the potential readers all useful information for an easier comprehension of whole content of the manuscript.

The study design is clear and well structured, integrating experimental methods (i.e., high resolution NMR) and computational methods (i.e., molecular dynamics), which provide interesting results that showed the role of lipid microdomain dynamics in APP processing.

The Authors described all  methods in detail, making the findings reproducible and interpretable. Results are well presented, and the figures make the complex data more accessible to the readers. All findings are contextualized in light of current literature  and the conclusions support obtained data.

The Authors could improve the manuscript by considering the application of the obtained findings into clinical practice through specific strategies able to avoid the pathological processing of APP.

Author Response

Thank you very much for your high rating of our work. Please find the detailed response to your comment below.  The corresponding correction is highlighted in the re-submitted manuscript.

Reviewer comment

The Authors could improve the manuscript by considering the application of the obtained findings into clinical practice through specific strategies able to avoid the pathological processing of APP.

Thank you for pointing this out. A general statement on a possible sterol-based therapeutic agent targeting the new putative binding site to modulate APP processing was added to the end of the Discussion (lines 420-423).

Reviewer 2 Report

Comments and Suggestions for Authors

This study examines the interaction of sterols, specifically a cholesterol analogue, with the transmembrane domain of amyloid precursor protein (APP) in different lipid environments. It uses NMR spectroscopy and molecular dynamics simulations to explore its role in Alzheimer's disease (AD) pathogenesis.

Strengths of the study

Innovative Techniques: Advanced NMR and simulations reveal new sterol-APP interaction sites, advancing knowledge of lipid-driven AD mechanisms.

Comprehensive Insights: Demonstrates the dynamic sterol-APP interactions across various membrane systems and their pathological relevance.

Therapeutic Implications: Highlights the lipid environment's role in AD, supporting potential membrane composition interventions.

Future suggestion:

Validate findings with in vivo studies.

Include a broader range of lipid species in the analysis.

Explore therapeutic strategies to modulate lipid environments and prevent pathological APP cleavage.

Author Response

Thank you very much for your careful review of our work. Please find the detailed response to your comment below.  The corresponding correction is highlighted in the re-submitted manuscript.

 Future suggestion:

Validate findings with in vivo studies.

Include a broader range of lipid species in the analysis.

Explore therapeutic strategies to modulate lipid environments and prevent pathological APP cleavage.

We thank the reviewer for valuable advice on further development of our work and gratefully accept his suggestions, including the proposals to confirm our results using in vivo studies, to expand the range of the assays, and to explore therapeutic strategies for modulating the lipid environment to prevent pathological cleavage of APP.

In addition, in the revised manuscript, we noted the possibility of modulating APP processing by a sterol-based therapeutic agent targeting the novel binding site identified in our work (lines 420–423).

Reviewer 3 Report

Comments and Suggestions for Authors

The article, "Diverse interactions of sterols with APP transmembrane domain can shift the distribution between the alternative amyloid-β production cascades in a manner dependent on local lipid environment," investigates the role of sterols, particularly cholesterol analogs, in modulating the cleavage pathways of the amyloid precursor protein (APP) transmembrane (TM) domain under different lipid membrane conditions. The authors use high-resolution NMR spectroscopy and molecular dynamics (MD) simulations to analyze the interactions between spin-labeled cholesterol (doxyl-cholestane) and the APP TM domain.

The integration of NMR spectroscopy and MD simulations provides a detailed and complementary understanding of sterol-APP interactions. Furthermore, high-resolution NMR data strengthens the reliability of the observed interaction sites.

The identification of the hinge region interaction site in disordered membranes is a notable advancement, expanding current knowledge of APP-cholesterol interactions. The study successfully links lipid membrane states (ordered vs. disordered) to APP conformational changes, which are relevant for secretase cleavage pathways.

The findings provide a mechanistic explanation for how lipid environment and cholesterol may influence Aβ production, which is central to AD pathology.

The manuscript is well-structured, with clear descriptions of the methods and results. Supplementary materials provide thorough supporting data, particularly the independent MD simulations for each sterol-binding site.

 Major Comments

The use of spin-labeled cholesterol (doxyl-cholestane) raises concerns about its physiological relevance compared to native cholesterol. The authors briefly discuss this limitation but could elaborate further on how the results align with native cholesterol behavior in biological membranes.

While the study highlights structural changes induced by cholesterol interactions, the mechanistic link to APP cleavage by α-, β-, and γ-secretases remains indirect. A more detailed discussion of how these structural perturbations influence secretase accessibility would enhance the biological relevance of the findings.

The MD simulations use POPC bilayers, but the authors do not discuss how other lipid species (e.g., sphingomyelin or gangliosides) might modulate cholesterol-APP interactions. A brief discussion of lipid composition variability in biological membranes would add context.

The study assumes a fixed cholesterol concentration in the membrane. Discussing how varying cholesterol concentrations (e.g., under physiological or pathological conditions) might influence APP-cholesterol interactions could provide additional context and relevance to the findings.

The findings highlight cholesterol-binding sites that modulate APP cleavage pathways, but the implications for therapeutic development are underexplored. Discussing how these insights could guide the design of small molecules or sterol derivatives to regulate Aβ production would increase the study’s translational value.

The manuscript could benefit from a more explicit comparison with prior research on APP-cholesterol interactions. Highlighting similarities and differences in identified binding sites or structural effects would help position this work within the broader literature.

Author Response

Thank you very much for taking the time to carefully review this manuscript. Below you will find Point-by-point responses to your comments and suggestions. Relevant corrections are highlighted in the resubmitted manuscript.

Comments and Suggestions for Authors

 1. The use of spin-labeled cholesterol (doxyl-cholestane) raises concerns about its physiological relevance compared to native cholesterol. The authors briefly discuss this limitation but could elaborate further on how the results align with native cholesterol behavior in biological membranes.

As it is explicitly mentioned in the text, doxyl-cholestane interaction sites of APP TM domain coincide with those of native cholesterol, implying that the spin label does not alter the interaction modes and cholestane adequately represents cholesterol behavior. Although cholestane is structurally more similar to oxydized forms of cholesterol, the native cholesterol can, in appropriate membrane environments, adopt tilted orientations preferred by cholestane and affect APP TM domain conformation and dimerization modes as described in the paper. A statement to that effect is added to discussion (lines 268-276).

2. While the study highlights structural changes induced by cholesterol interactions, the mechanistic link to APP cleavage by α-, β-, and γ-secretases remains indirect. A more detailed discussion of how these structural perturbations influence secretase accessibility would enhance the biological relevance of the findings.

A brief statement on secretase recognition and cleavage depending on cholesterol interactions was added to discussion (lines 414-418). Any attempts to elaborate further on the subject matter appears to lead to unsubstantiated speculations since it requires further focused investigation due to highly heterogenous nature of the system.

3. The MD simulations use POPC bilayers, but the authors do not discuss how other lipid species (e.g., sphingomyelin or gangliosides) might modulate cholesterol-APP interactions. A brief discussion of lipid composition variability in biological membranes would add context.

A brief statement on the effect of lipid composition and phase state on cholesterol-APP interaction was added (lines 279-288).

4. The study assumes a fixed cholesterol concentration in the membrane. Discussing how varying cholesterol concentrations (e.g., under physiological or pathological conditions) might influence APP-cholesterol interactions could provide additional context and relevance to the findings.

The issue, however interesting, appears to be beyond the scope of the present paper. Though it is generally apparent that the concentrations of cholesterol and its derivatives would affect their interactions with APP, the complicated homeostasis of these molecules prevents any credible assumption on such effects. Consideration of dependence of cholesterol-APP interactions on cholesterol concentration would require a profound review and analysis since besides direct concentration effects, different oligomerization of APP and phase state of the lipid environment has to be considered (lines 371-376).   However, a brief discussion on the issue was added (lines 400-405)  

5. The findings highlight cholesterol-binding sites that modulate APP cleavage pathways, but the implications for therapeutic development are underexplored. Discussing how these insights could guide the design of small molecules or sterol derivatives to regulate Aβ production would increase the study’s translational value.

A general statement on a possible sterol-based therapeutic agent targeting the new putative binding site to modulate APP processing was added to the end of the Discussion (lines 418-423).

6. The manuscript could benefit from a more explicit comparison with prior research on APP-cholesterol interactions. Highlighting similarities and differences in identified binding sites or structural effects would help position this work within the broader literature.

Explicit correlation between the interaction sites observed in prior works and those we determined in the present paper have been added throughout the Discussion (by means of specifying the numbers corresponding to the sites in our nomenclature).

Round 2

Reviewer 3 Report

Comments and Suggestions for Authors

I confirm that the authors have thoroughly addressed all of my suggestions and questions. I find their responses satisfactory, and the manuscript meets the standards for publication. I recommend its acceptance.